# Strategies for Proteome-Wide Quantification of Glycosylation Macro- and Micro-Heterogeneity

**DOI:** 10.3390/ijms23031609

**Published:** 2022-01-30

**Authors:** Pan Fang, Yanlong Ji, Thomas Oellerich, Henning Urlaub, Kuan-Ting Pan

**Affiliations:** 1Department of Biochemistry and Molecular Biology, School of Biology & Basic Medical Sciences, Suzhou Medical College of Soochow University, Suzhou 215123, China; pfang@suda.edu.cn; 2Bioanalytical Mass Spectrometry Group, Max Planck Institute for Multidisciplinary Sciences, 37077 Göttingen, Germany; yanlong.ji@mpibpc.mpg.de; 3Hematology/Oncology, Department of Medicine II, Johann Wolfgang Goethe University, 60590 Frankfurt am Main, Germany; thomas.oellerich@kgu.de; 4Frankfurt Cancer Institute, Johann Wolfgang Goethe University, 60596 Frankfurt am Main, Germany; 5German Cancer Consortium (DKTK), Partner Site Frankfurt/Mainz, German Cancer Research Center (DKFZ), 69120 Heidelberg, Germany; 6Institute of Clinical Chemistry, University Medical Center Göttingen, 37075 Göttingen, Germany

**Keywords:** glycoproteomics, quantification, mass spectrometry, stable-isotope labeling, label free

## Abstract

Protein glycosylation governs key physiological and pathological processes in human cells. Aberrant glycosylation is thus closely associated with disease progression. Mass spectrometry (MS)-based glycoproteomics has emerged as an indispensable tool for investigating glycosylation changes in biological samples with high sensitivity. Following rapid improvements in methodologies for reliable intact glycopeptide identification, site-specific quantification of glycopeptide macro- and micro-heterogeneity at the proteome scale has become an urgent need for exploring glycosylation regulations. Here, we summarize recent advances in N- and O-linked glycoproteomic quantification strategies and discuss their limitations. We further describe a strategy to propagate MS data for multilayered glycopeptide quantification, enabling a more comprehensive examination of global and site-specific glycosylation changes. Altogether, we show how quantitative glycoproteomics methods explore glycosylation regulation in human diseases and promote the discovery of biomarkers and therapeutic targets.

## 1. Introduction

Glycosylation occurs ubiquitously on vital biomacromolecules, including proteins, lipids and nucleic acids [1]. In particular, protein glycosylation, one of the most common and diverse post-translational modifications (PTMs) of proteins, decorates the side chains of the acceptor amino acid residues, most commonly by N- and O-glycosidic linkages (Figure 1A). Protein glycosylation plays critical roles in various biological processes, and abnormal glycosylation is often linked with disease states [2]. Recent studies have further revealed the correlation between glycosylation and molecular subtypes of various cancers, suggesting that glycosylation profiling has the potential to stratify patients for precision treatment [3,4,5]. Along with urgent analytical needs, mass spectrometry (MS) has emerged as a powerful tool for global glycosylation profiling. Continuous developments in MS technologies, glycopeptide enrichment, and database search algorithms have significantly improved the coverage and reliability of the identification of intact glycopeptides in complex biological samples [6,7], omitting the need for separate analyses of chemically or enzymatically released oligosaccharides (glycome) and de-glycosylated peptides (de-glycoproteome) [8]. Such improvements in intact glycopeptide analysis preserve the information of protein-glycan linkages and increase the range of applications of glycoproteomics to complex biological samples. Despite this progress, the robust and reliable quantification of intact glycopeptides at the proteome scale remains technically challenging, owing mainly to the complexity and the macro- and micro-heterogeneity of site-specific glycosylation. Of which, macro-heterogeneity refers to the glycosylation occupancy of each glycosite (i.e., the degree of the site being glycosylated on the protein molecules), whereas micro-heterogeneity describes the relative abundances of all identifiable glycoforms on the occupied glycosite (Figure 1B). In contrast to the quantification of other PTMs (e.g., phosphorylation), where one determines either relative modification levels or occupancies of the modified sites in different samples, quantitative glycoproteomics additionally measures site-specific micro-heterogeneity. Among the various methodologies of protein glycosylation analysis, both intact glycopeptide and de-glycosylated peptide analyses can resolve glycosylation macro-heterogeneity, but only intact glycopeptide analysis allows for micro-heterogeneity determination (Figure 1C). Of note, despite the recent developments in MS-based glycoproteomics for more detailed glycan structure examination [9,10], fully resolved glycan linkages are still only available via glycan release and subsequent glycomic analysis. We discuss different levels of glycopeptide quantification in more detail in a separate section below.

This article summarizes current MS-based technologies for proteome-wide quantification of N- and O-linked protein glycosylation (Table 1). These technologies are mainly derived from the principles of quantitative proteomics approaches that have been nicely reviewed elsewhere [11,12], such as isotopic/isobaric chemical labeling and metabolic labeling. We thus stress more on the adaptations and applications of the methods for quantitative glycoproteomics without repeatedly detailing the analytical theory of the methods. Next, we describe a data-propagation strategy to achieve multi-layered quantitative glycoproteomics, reporting comprehensive glycosylation changes at the glycosite (macro-heterogeneity), glycoform (micro-heterogeneity) and glycan (glycan composition) levels. Finally, we summarize recent glycoproteomics studies in human disease and highlight the importance of quantitative glycoproteomics in biomarker and therapeutic target discovery.

## 2. Labeling-Based Quantification

Following the developments in proteomics, quantitative glycoproteomics approaches commonly use stable isotopes, such as ^13^C, ^15^N, ^18^O and ^2^H, to label glycopeptides. Such labeling creates mass shifts to glycopeptide precursors or fragmentation products so that glycopeptides from different samples can be differently labeled and pooled before measurement but remain distinguishable by mass spectrometry. Stable isotopes can be incorporated chemically, metabolically or enzymatically into amino acids or glycans on glycopeptides (Figure 2). These strategies can be further classified as isotopic or isobaric labeling, where the relative quantification of the labeled glycopeptides is achieved at either MS1 or MSn levels, respectively.

### 2.1. Isobaric Chemical Labeling

Isobaric labeling reagents, such as tandem mass tags (TMT), isobaric tags for relative and absolute quantitation (iTRAQ), isotope-encoded dimethylated leucine or isobaric N, N-dimethyl leucine (DiLeu) and isobaric tag (IBT), permit global glycopeptide quantification with high sample multiplexity (up to 18 parallel samples for TMT and 21 for DiLeu) [13,74,75,76]. These reagents commonly consist of an amine-reactive group (reacting with the glycopeptide’s N-terminus and with the side chain of lysine), a mass balancer and a reporter. Glycopeptides from different samples are labeled after proteolysis with a series of isobaric reagents separately, followed by sample pooling and enrichment (see Figure 2 workflow). During LC-MS analysis, differentially labeled glycopeptides coelute chromatographically and appear as one peak in MS1 scans. However, upon MSn fragmentation, the isobaric glycopeptide ions from different samples generate a series of reporter ions with distinct masses, enabling relative quantification of the glycopeptides in the samples to be compared. These isobaric labeling reagents were originally designed for general proteomic analysis. In most cases, the same labeling procedure used for standard peptides is readily applicable to quantitative glycoproteomics, although modifications in the labeling procedure (such as omitting acetone precipitation, lowering cysteine protecting reagent concentration, and increasing triethylammonium bicarbonate buffer concentration) were noted for improved glycopeptide labeling efficiency [14].

Rapidly improving software packages further speeds up the applications of isobaric labeling to quantitative glycoproteomics. For example, Byonic, one of the pioneering software programs developed for large-scale identification of both N- and O-linked glycopeptides [77], can work as an embedded node in the quantitative workflows of another commercial software, Proteome Discoverer (PD). This platform has emerged as the most widely used database search algorithm in the field [78]. Its applications include the site-specific quantification of N- and O-linked glycosylation on protein therapeutics [15], urinary N-glycoprotein profiling in prostate cancer patients [16] and the determination of N-glycoproteome dynamics associated with prostate cancer progression [17]. In addition, the Parker group integrated quantitative glycomics and glycoproteomics to reveal the functional roles of N-glycosylation in myogenesis and muscle development [18]. The Lu group also used iTRAQ and Byonic workflows to identify glycopeptides of serum mannose receptors as potential biomarkers to differentiate the subtypes of breast cancer [19].

GPQuest, developed by the Zhang group, also allows for confident identification and isobaric-labeling-based quantification of intact N-glycopeptides [3,4,20,21,22,23,79]. In addition to intact glycopeptide analysis, the group simultaneously performed proteome and de-glycoproteome quantification to determine the site-specific micro- and macro-heterogeneity of N-glycosylation [3,24]. Using their workflow, Zhang and her coworkers successfully determined site-specific glycosylation changes in numerous human disease models and patient samples, including prostate cancer cells, ovarian carcinoma cell lines, heart tissues of mice with transverse aortic constriction induced cardiac hypertrophy, aggressive prostate cancer cell lines, luminal and basal-like types of breast cancer patient-derived xenografts (PDXs) and hepatocellular carcinoma tumors [20,21,22,23,24,80]. The Zhang group further applied their workflow to characterize clinical samples and demonstrated the method’s potential for biomarker discovery and cancer subtype stratification, an important step toward precision treatment [3,4].

Stadlmann et al. also developed a comparative glycoproteomics platform combining a new N-glycopeptide identification algorithm, SugarQb, with the quantitative workflows embedded in Proteome Discoverer [25]. They used this method to explore the micro-heterogeneity of global N-glycosylation in human and mouse embryonic stem cells, leading to the identification of new players in ricin toxicity. This platform also allowed them to determine PNGase-F-resistant glycopeptides and reveal tryptic substrate specificities of PNGase-F [26].

Multiplexed isobaric labeling reduces the overall LC-MS measurement time and variations introduced between measurements [74,75]. In addition, by pooling all samples together, it boosts the signal of low-abundance species that are otherwise not detectable in any individual sample. Isobaric tags themselves also increase the ionization efficiencies of peptides or glycans [81]. Since the majority of the glycoforms are present with low stoichiometry, multiplexing enhances the sensitivity and the depth of site-specific glycoproteomics. Despite the advantages, standard MS2-based methods used for multiplexed samples often suffer from impaired quantification accuracy caused by co-isolation interference [82]. To solve these limitations, we recently introduced a multi-notch MS3 method (Glyco-SPS-MS3) for quantifying TMT-labeled glycopeptides with higher accuracy [27]. The optimized Glyco-SPS-MS3 improved both the identification and quantification of TMT-labeled N-glycopeptides in complex biological samples. Such improvements can be further enhanced by implementing ion-mobility spectrometry in the LC-MS analysis [28]. Of note, raw files generated by Glyco-SPS-MS3 are not readily acceptable by conventional glycopeptide search engines. We thus developed GlycoBinder [27] to merge MS2 and MS3 spectra into pseudo-fragment ion spectra. Subsequently, GlycoBinder calls pGlyco2 [44] for glycopeptide identification and RawTools [83] for reporter ion intensity extraction. The pGlyco 2 algorithm determines false discovery rates (FDR) at both glycan and peptide levels, enhancing the reliability of glycopeptide identification. Although this software allows identifying labeled glycopeptides, it does not support labeling-based quantification in its native format. One has to employ another quantification tool, such as pQuant [84] or GlycoBinder, to extract ion intensities of identified glycopeptides for relative quantification between labeling channels.

Along the same line, Zhu et al. combined electron-transfer dissociation (ETD) with a standard multi-notch workflow, resulting in improved identification and quantification of TMT-labeled intact glycopeptides [85]. Using an Orbitrap Tribrid mass spectrometer, the method starts with a common synchronous precursor selection (SPS) workflow, where the selected parent ion is fragmented with collision-induced dissociation (CID) in the ion trap, and the top 10 MS2 fragment ions are further isolated and fragmented using higher-energy collisional dissociation (HCD) with the resulting MS3 fragments detected in the Orbitrap. In addition, a separate electron-transfer dissociation (ETD) MS2 is performed on the same parent ion in the ion trap. The combination of CID and ETD spectra ensures good glycopeptide identification confidence, and the SPS-MS3 quantification improves the quantification accuracy and precision.

In summary, rapidly developing MS technologies and software tools enhance the quality of intact glycopeptide analysis and enable more straightforward data analysis for non-experts. Although commercial isobaric labeling reagents may not be affordable to all due to the expensive costs, recent developments have significantly promoted their applications in quantitative glycoproteomics.

### 2.2. Isotopic Chemical Labeling

In contrast to isobaric labeling, isotopic chemical labeling allows the quantification of intact N-/O-glycopeptides at the MS1 level. Its multiplexing capacity is often limited to up to three (light, medium, and heavy) due to the increased MS1 spectrum complexity and overlapping glycopeptide isotopic peaks. Labeling is commonly performed after proteolysis and followed by equal mixing. After enrichment, differently labeled glycopeptides co-elute chromatographically and appeared distinguishable in the MS1 spectra. The relative peak intensities/areas of the light and heavy-labeled glycopeptides are compared.

Dimethyl labeling is based on the reaction of peptide primary amines (N-terminus or ϵ-amino group of Lys residue of a peptide) with formaldehyde, introducing 4n (^12^CD^2^O) or 6n Da (^13^CD^2^O) mass difference, where n is the number of amino groups in peptides. The Tian group developed a dedicated software GPSeeker and GPSeekerQuan for the identification and quantification of dimethyl labeled intact N-glycopeptides [9]. Based on the identified intact glycopeptide sequences, the software calculates the theoretical mass shift caused by isotopic labeling and looks for the paired precursor isotopic peaks in the MS1 spectra. Each pair of the isotopic envelopes requires three isotopic peaks per label for successful quantification. Peak intensities of the three isotopic peaks in each isotopic envelope were summed to deduce the abundance ratio of the glycopeptide pair. The Tian group further applied stable isotopic diethyl labeling (SIDE), where N-glycopeptides were diethylated with deuterium-free reagents (XH_3_XHO, X = ^13^C or ^12^C) and NaBH_3_CN, for the relative quantitation of intact N-glycopeptides with enhanced accuracy and dynamic range [29]. Using these quantitative glycoproteomics methods, the group successfully identified aberrant N-glycosylation in gastric cancer tissues, cell-surface N-glycoprotein markers of MCF-7/ADR cancer stem cells and potential N-glycosylation markers for pancreatic cancer [30,31,32,33,34,35].

Byonic and PD are also compatible with dimethyl-labeling-based intact N- and O-glycopeptide quantification [36,37,38]. Schjoldager et al. combined dimethyl labeling with the SimpleCell strategy for quantitative O-glycoproteome analyses of HepG2 cell lines with GalNAc-T isoforms mutated and explored how the GalNAc-T isoform repertoire quantitatively affects the O-glycoproteome [37,86]. With the same strategy, they further quantified the O-glycoproteome of a panel of isogenic HEK293 cells with various knockouts of GalNAc-T genes (GALNT1, T2, T3, T7, T10 or T11), leading to the identification of O-glycosylation sites regulated by specific GalNAc-T isoforms [38].

Zhang et al. used differential glycan oxidation and dimethylation-based quantitative de-glycopeptide analysis to determine N-sialoglycan occupancy rates on glycoproteins [39]. The method is based on the fact that mild periodate concentrations can specifically oxidize sialic acid moieties on glycoproteins to aldehydes, while high periodate concentrations oxidize all types of glycans. After differential oxidation of the glycans, glycoproteins were captured using hydrazide chemistry, and differentially oxidized glycopeptides were subsequently labeled with heavy and light dimethyl reagents, respectively, and released by PNGase F treatment. The resulting heavy-to-light ratio of released de-glycopeptides thus represents the N-sialoglycan occupancy rates. The group used this approach to determine the N-glycosylated sites showing significant changes in sialylation occupancy rates between hepatocellular carcinoma and normal human liver tissues.

Compared with commercial isobaric reagents, the above isotopic chemical labeling approaches are efficient and have lower costs. However, some reactions may give rise to side products [87]. In addition, the lower multiplexing capacity further limits its applications with larger sample amounts.

### 2.3. Metabolic Labeling

The stable isotope labeling by amino acids in cell culture (SILAC) method incorporates ^13^C- or ^15^N-labeled amino acids (“heavy” amino acids, often with lysine and arginine) into endogenous proteins metabolically during cell culture [88]. Differently labeled proteins extracted from up to three conditions are mixed immediately after cell lysis, minimizing potential system errors introduced during sample processing. Heavy glycopeptides and their light counterparts co-elute chromatographically and are shown as duplets with predefined mass differences in MS1 scans. Relative quantification is then achieved by comparing the intensities of light and heavy peaks. Following the broad applications in proteomics, the SILAC approach is also adapted for glycopeptide quantification.

Benjamin et al. performed quantitative N-glycomics and SILAC-based quantitative N-glycoproteomics to study protein glycosylation changes in adipocytes upon TNF-alpha-induced insulin resistance [40]. They observed increased terminal di-galactose and decreased biantennary alpha-2,3-sialoglycans in TNF-alpha-treated 3T3-L1 adipocytes, which correlated well with the upregulation of B4GalT5 and Ggta1 galactosyltransferases and the downregulation of ST3Gal6 sialyltransferase, as determined by their proteome profiling. They then used 2-plex SILAC (Arg0/Lys0 or Arg10/Lys8) to label 3T3-L1 adipocytes and performed SILAC-based quantification on enriched N-glycopeptides with and without deglycosylation to determine N-glycosylation occupancy and site-specific N-glycosylation regulation, respectively. The identification and quantification of intact glycopeptides was performed with Proteome Discoverer using the Byonic node. By normalizing to protein abundance changes, they found that the relative N-glycan occupancy remained largely unchanged. In addition, only 16 out of the 56 TNF-alpha regulated site-specific N-glycopeptides showed protein-levelindependent alterations, suggesting site-specific changes in glycosylation. This study highlighted the importance of multi-layered (at the layers of protein abundance, occupancy and site-specific glycoform change) quantification when assessing N-glycosylation regulations.

Although SILAC reduced quantitation errors introduced during sample preparation, it is not applicable to many biological materials, especially to patient tissues, and has a limited number (usually up to three) of conditions to be compared in one experiment. Micro-heterogeneous glycoforms on one peptide core have closely related masses and do not separate well on the standard C18 chromatography, which often makes MS1 spectra of intact glycopeptide analysis more complicated. SILAC inherently further increases the MS1 complexity, resulting in interfered SILAC pair determination and chromatographic ion extraction. The introduced SILAC pair can also cause the over-sequencing of the same glycopeptides and undersampling of all available glycopeptides in DDA analysis.

### 2.4. Enzymatic Labeling Using ^18^O Stable Isotope

PNGase F efficiently releases N-glycans and introduces one ^18^O to the former N-glycosylated asparagine residues when reacting in an ^18^O-labeled buffer. The +2 Da mass shift enables glycosite identification and glycosylation occupancy determination [89,90,91]. To improve the quantification accuracy impaired by the partially overlapped isotopic distribution of the ^16^O- and ^18^O-labeled peptides, Liu et al. developed tandem ^18^O stable isotope labeling (TOSIL), in which three ^18^O atoms are incorporated into de-glycosylated peptides for determining N-glycosylation site occupancy [92]. The TOSIL method introduces two ^18^O atoms into the carboxyl termini of all peptides during tryptic digestion and a third ^18^O atom into the N-glycosylation site while cleaving glycans by PNGase F. Thus, compared to normal ^16^O de-glycopeptides, a mass shift of 6 Da for singly glycosylated peptides or 8 Da for doubly glycosylated peptides appears in MS1 scans. The intensity ratio of ^18^O/^16^O isotopic peaks represents the abundance difference between the glycosylated peptide (including all glycoforms) and its non-glycosylated counterpart, allowing for determining changes in N-glycosylation site occupancy [41,42].

Recently, Zhang et al. further applied the ^18^O labeling methods for the relative quantification of intact glycopeptides in serum from patients with hepatitis-B-virus-related liver diseases [43]. They introduced two ^18^O atoms into the carboxyl termini of intact glycopeptides during tryptic digestion, resulting in a +4 Da mass difference in mass spectrometry. A combination of pGlyco [44] and pQuant [84] was used for intact glycopeptide identification and quantification.

Enzymatic labeling is a relatively simple and cost-effective approach for quantitative glycoproteomics. However, downstream data analysis should consider the back exchange of ^18^O to ^16^O during sample preparation. In addition, the overlapped isotopic peaks of ^18^O- and ^16^O-labeled glycopeptides (with only a 4 Da mass difference) also complicate the data processing steps for quantification.

### 2.5. Glycan Labeling

In addition to the peptide backbones, stable isotopes can also be incorporated into glycans attached to glycopeptides. The isotopic methylamine (MeSIL) method labels the carboxyl groups on both the sialic acid residues and the peptide moiety (such as Asp, Glu and C-terminus) at the same time [45]. Labeled glycopeptides show a 3*N Da mass shift (N represents the number of labeled carboxyl groups) in MS1 scans. As a result, after mixing differentially labeled samples, the isotopic pairs of labeled glycopeptides allow for the relative quantification of intact glycopeptides between samples. The labeled sialic acids and the resulting mass shifts also reduce the likelihood of miss-assigned glycan composition.

Alternatively, metabolic labeling using ^15^N-enriched or ^13^C-enriched media can replace all the N and C atoms in glycopeptides with ^15^N and ^13^C, respectively. This method was successfully applied to yeast N-glycoproteomics. Liu et al. applied the ^15^N and ^13^C labeling to yeast glycopeptides to evaluate the true false discovery rate (FDR) of pGlyco [44]. Equal amounts of unlabeled and ^15^N- or ^13^C-labeled yeast proteins were pooled and then analyzed in one LC-MS/MS run. The detected mass difference between an unlabeled glycopeptide and its ^15^N or ^13^C-labeled counterpart should be equal to the number of nitrogen or carbon atoms calculated based on the assigned glycopeptide composition. Accordingly, they filtered out the false identifications and determined the true FDR. To the best of our knowledge, pQuant is the only software supporting quantifying glycopeptides bearing ^15^ N- and ^13^C-labeled glycans.

Biocompatible azido-bearing monosaccharides are commonly used for metabolic glycan labeling [93,94]. The Wu group combined azido metabolic labeling and click chemistry to enrich and map sialoglycosylated proteins with glycans cleaved on the cell surface [95]. Based on SILAC-based quantification, they determined the differences in cell surface N-sialoglycoproteins in invasive MDA-MB-231 and non-invasive MCF-7 breast cancer cells. They extended the method to label the entire surface of N-glycoproteins using various azido-bearing analogs and to quantify surface N-glycosylation changes in statin-treated liver cells [96]. Intriguingly, the Wu group further integrated their approach with pulse-chase metabolic labeling and TMT/SILAC-based quantification to systematically analyze surface glycoprotein dynamics and determine their degradation and synthesis rates [97,98].

## 3. Label-Free Quantification

Label-free quantification (LFQ) methods aim at quantifying glycopeptides without the use of stable-isotope labels. The samples to be compared were prepared separately, including protein extraction, digestion, glycopeptide enrichment and parallel LC-MS/MS measurements. The extracted ion currents (XIC) or spectral counts of identified glycopeptides across runs were compared. Depending on the MS acquisition methods, the LFQ can be classified into data-dependent acquisition (DDA) and data-independent acquisition (DIA) based quantification (Figure 3). In the DDA mode, the most abundant precursor ions detected in an MS1 survey scan are selected and isolated sequentially with a narrow mass-to-charge (m/z) window (usually 0.5–2 Th) for MS2 fragmentation. The number of selected precursors per acquisition cycle is based on the survey scans and pre-defined settings. In the DIA mode, however, precursor isolation ranges (usually > 10 Th), the number of MS2 events and precursor m/z coverage are pre-defined. Each acquisition cycle repeats the same scan events accordingly, independent of the data acquired.

### 3.1. DDA-Based Label Free Quantification

DDA-based label-free quantification uses either the MS1 intensities or the spectral counts of identified glycopeptides for simple and accurate glycopeptide quantification. Benefitting from the rapidly developing software packages, numerous data processing tools support intact glycopeptide LFQ analysis (with or without an independent intact glycopeptide identification algorithm), including LFQuant [46], iBAQ [47], Byologic [99], PD [48], Xtractor 2D [49], Mascot Distiller [50] and XIC using Xcalibur [51]. Alternatively, one can extract the quantification information of identified glycopeptide-to-spectrum matches via Skyline [52] or from the MaxQuant result file “allpeptide.txt” (based on their MS/MS scan numbers) [53].

Glycosylation heterogeneity makes the LFQ of intact glycopeptides more troublesome than the standard proteomic analysis. Glycoforms sharing the same peptide sequence can have close masses and elute closely in a narrow retention time range, resulting in overlapping glycopeptide signals and thus interfering with glycopeptide peak extraction. Nevertheless, by combining the intensities of the top three isotope peaks at the three highest MS spectral points, Integrated GlycoProteome Analyzer (I-GPA) enables fast and sensitive glycopeptide LFQ [54]. Although LFQ demands highly reproducible LC-MS performance across all runs, an interlaboratory study still showed convincing quantification reproducibility of <25% coefficient of variation among replicate runs performed in four different laboratories [100]. These studies demonstrated that, despite the complexity of glycosylation, the DDA-based glycopeptide LFQ is robust and reproducible.

DDA analysis is inherently biased against low-abundance glycopeptides, which either do not produce sufficient detectable fragment ions for reliable identification or are not selected for fragmentation at all. Limited by the scan speed of available mass spectrometers, it is common that many existing glycopeptides are not identified in all DDA runs, resulting in missing values for LFQ. Although numerous existing proteomics data processing methods allow assigning and matching MS1 features across all raw files to minimize LFQ missing values [101,102,103], they are not readily applicable to glycoproteomics. Zhao et al. developed an MS1 feature-based matching for intact O-glycopeptide quantification [55]. They determined a reference/calibrated retention time (RT) for each O-glycopeptide by taking the median RT of each O-glycopeptide with MS2 identifications across all measured runs and the average RT shift of all the O-glycopeptides in each run. They then transferred the sequence information across related LC-MS runs from identified O-glycopeptides to unidentified MS1 features with matched mass, charge, RT and isotopic patterns, achieving a 30% reduction in missing values and improved reproducibility.

An alternative method to MS-intensity-based DDA quantification is spectral counting, which utilizes counts of identified glycopeptide-to-spectrum matches (GPSMs) to represent the abundance of glycopeptides. Because of the random-selection and intense-peak-first triggering strategy of a DDA analysis, more abundant glycopeptides in theory lead to higher numbers of GPSMs than low-abundance peptides. This method was recently applied to quantify the site-specific glycosylation of SARS-CoV-2 spike protein and human ACE2 receptor [104]. Additionally, Yang et al. employed a spectral counting approach in their chemoenzymatic method for site-specific extraction and quantification of O-linked glycopeptides and determined changed O-glycoproteome in tumorous kidneys compared to normal tissues [57]. However, the number of GPSMs for a given glycopeptide depends on a number of factors, including the stochastic DDA sampling, dynamic exclusion setting, and the complexity of the sample matrix, making spectral count-based quantification less robust and reliable, especially for the glycopeptides with only a few GPSM counts.

### 3.2. DIA-Based Label-Free Quantification

With faster mass spectrometers and advanced data processing tools, data-independent acquisition (DIA) methods offering improved sensitivity and reproducibility compared traditional DDA are rising in quantitative proteomics [105]. This approach has also been applied for analyzing de-glycosylated peptides and intact N- and O-linked glycopeptides. The Aebersold group extended their SWATH-MS method to quantify PNGase F-treated de-glycopeptides in human plasma [59]. They systematically compared the performance of SWATH and selected reaction monitoring (SRM)-based quantification, concluding that SWATH resulted in a similar performance in variability, accuracy and dynamic range with a slightly lower sensitivity but much deeper glycoproteome coverage. The Schulz group also applied a similar SWATH method to determine the site-specific N-glycosylation occupancy of several N-glycosylation sites in N-glycoproteins from the yeast cell wall and from human saliva by comparing the intensities of de-glycosylated peptides and the total intensities of all peptides from the corresponding proteins [60]. This group further developed a targeted data-independent acquisition strategy termed SWAT (sequential window acquisition of targeted fragment ions) [106]. Unlike the standard SWATH method, which isolates precursors with a larger m/z window, SWAT only isolates selected peptides of interest with a 4 m/z window. They showed that SWAT provided robust occupancy measurements at N-glycosylation sites and with higher precision than SWATH, allowing identification of novel glycosylation sites dependent on the Ost3p and Ost6p regulatory subunits of oligosaccharyltransferase. More recently, Yang et al. reported an in-depth measurement of N-glycosylation stoichiometry changes caused by tunicamycin in human HEK-293 cells and by a temperature shift in Chinese hamster ovary cells [61]. To achieve a high glycoproteome coverage, they established a de-glycopeptide spectral library by analyzing prefractionated, lectin-enriched and PNGase F-treated de-glycopeptides from HEK-293 cells using DDA analysis. In total, they determined the stoichiometries of 2274 glycosites by comparing the intensities of de-glycosylated peptides and non-glycosylated peptides from different runs.

A DIA analysis of intact N-glycopeptides is much more complex than analyzing de-glycopeptides, requiring certain adjustments and the optimization on LC-MS settings (e.g., m/z ranges) and on data analysis pipelines (e.g., construction of glycopeptide spectral libraries). As one of the pioneers, Zacchi et al. applied DIA-based N-glycoproteomics to study the changes in glycan macro- and micro-heterogeneity in mature proteins caused by mutations in the N-glycosylation pathway in Saccharomyces cerevisiae [62]. They created the glycopeptide spectral library by selecting experimentally determined b- and y-ions that did not contain the glycosylated Asn residue. The DIA method considered N-glycan structures ranging from GlcNAc2 to Man15GlcNAc2 for glycosylation micro-heterogeneity determination. Although beam-type CID of glycopeptides predominantly generates glycan fragments with limited b- and y- ions, their SWATH method still allows the quantification of glycopeptides, corresponding to eight glycosylation sites in the yeast cells.

Similarly, Sanda and Goldman also reported a SWATH workflow for detecting IgG glycoforms from human plasma. However, instead of b- and y-ions, they used Y-ions generated under lower collision energy for quantification. These manually curated Y-ions with a high yield of up to 60% precursor intensity were proven to be highly specific to each glycoform [63]. In a follow-up study, they constructed a spectral library containing 161 glycoforms of 25 peptides from 14 protein groups, with which they detected 10 of 14 glycoproteins without any glycopeptide enrichment, revealing glycosylation changes between cirrhotic patients and healthy controls [64]. In parallel and independently, we and others further applied various DIA methods using different mass spectrometers for targeted intact glycopeptide analysis [65,107]. These studies demonstrated improved sensitivity of DIA analysis of intact glycopeptides in complex samples compared with the standard DDA analysis. DIA methods further showed the potential to discover unknown and/or undefined modifications on glycopeptides that were not identifiable in the standard DDA analysis.

More recently, Ye et al. developed a DIA-based strategy for quantitative O-GalNAc–type glycoproteomic analysis in complex biological samples [66]. They first analyzed the O-glycoproteome of human serum and their SimpleCell cell lines [86], which produce homogeneous HexNAc (Tn-) and Hex-HexNAc (T-) O-glycans using DDA analyses. They then included all the identified glycopeptides to generate a combined Tn-/T-glycopeptide library containing more than 2000 glycoproteins with more than 11,000 unique glycopeptide sequences. The comprehensive O-glycopeptide library substantially improved the O-glycoproteome coverage. They showed a further improved O-glycopeptide identification by the in silico addition of NeuAc–HexNAc (STn), NeuAc–Hex–HexNAc (ST) and NeuAc2–Hex–HexNAc (diST) epitopes, as well as the non-glycosylated forms, into the Tn-/T-glycopeptide library.

In general, the confident identification of glycopeptides requires several levels of information: peptide sequence identification, glycan moiety identification and, ideally, glycosite localization. Numerous studies have elucidated strategies to estimate the false discovery rate (FDR) of glycopeptides in a DDA analysis [7]. However, FDR control of glycopeptides in DIA analysis is not yet mature. Yang et al. recently proposed GproDIA [67], a framework for the proteome-wide characterization of intact glycopeptides from DIA data with comprehensive statistical control using a 2-dimensional FDR approach and a glycoform inference algorithm. The method enables the accurate identification of intact glycopeptides in DIA analyses, even with wider isolation windows. GproDIA showed superior data completeness of glycopeptide identification and quantitative accuracy and precision compared with the standard DDA methods.

In summary, DDA-based glycopeptide LFQ, either using MS1 feature intensity or spectral count, has the advantage of simple workflows and lower cost but requires sophisticated data processing methods, retention time alignment, and post-acquisition normalization to account for the MS response variations in replicate measurements. This approach also suffers from severe missing values in large-scale glycoproteomics due to drastic differences in glycoform abundances, and low identification rates of less abundant glycopeptides in DDA. DIA-based LFQ methods have shown the potential to quantitate intact glycopeptides with higher sensitivity and fewer missing values. However, current glycopeptide DIA analysis still relies on a pre-established, DDA-generated spectral library, limiting its application. The so-called directDIA approach and the use of a predicted spectral library are not yet available in the glycoproteome field [108]. Nevertheless, DIA can significantly increase the identification rate of low-abundance glycopeptides. This should resolve the issue of stochastic sampling in DDA mode and increase the detection sensitivity level and quantification accuracy.

## 4. Target Analysis Using SRM/MRM

In targeted acquisition methods, such as selected or multiple reaction monitoring (SRM or MRM) or parallel reaction monitoring (PRM), LC-MS assays are deployed to detect glycopeptides of interest with high sensitivity, reproducibility and quantitative accuracy. SRM analysis was initially developed on triple quadrupole instruments, where the first quadrupole (Q1) scans for a pre-defined precursor, which undergoes CID fragmentation in the second quadrupole (Q2). Subsequently, pre-defined ions among the resulting fragments are scanned in the third quadrupole (Q3) (Figure 4A).

To achieve robust and sensitive glycopeptide detection in an SRM/MRM analysis, transition (i.e., a predefined pair of precursor and fragment ions that represent the quantity of the target glycopeptide) selection is crucial. Depending on the fragmentation modes, various types of glycopeptide fragments can be selected as target transitions, including B ions (oxonium ions), Y ions (intact peptide sequences with partial glycans) and b/y ions (peptide fragment ions without glycans) (Figure 4B). Monitoring the oxonium ions in a targeted glycopeptide analysis offers better sensitivity with less selectivity. This approach has been used to confirm the quantitative differences of glycopeptides from patient serum associated with esophageal disease [68] and quantify the fucosylated fraction of hemopexin and complement factor H in the plasma of patients with liver disease [69] and plasma immunoglobulins in cirrhosis (CIR) and hepatocellular carcinoma (HCC) [109]. As all glycopeptides generate common oxonium ions, such an assay can be interfered with when analyzing complex samples. The combination of multiple fragment ion types (b- and y-ions, oxonium ions and Y ions) increases selectivity and decreases the rate of false identifications. Using PRM assays, which detect all MS2 fragment ions of the target precursor, researchers determined the glycosylation changes of protein biomarkers for hepatocellular carcinoma [70,71] and measured PD-1, PD-L1 and PD-L2 at fmol/µg protein levels from melanoma biopsies [72].

The major drawback of targeted acquisition is that the number of precursor ions to be monitored is limited by the scan speed of the mass spectrometer. N-linked glycopeptides typically display heterogeneous glycoforms that elute within a narrow retention time window, resulting in overlapping LC peaks. Given that robust quantification requires at least 6–8 data points across a chromatographic peak, glycosylation heterogeneity often overwhelms the MS instrument duty cycle.

## 5. Multi-Layered Quantification of Glycoproteome

As mentioned above, various quantitative glycoproteomics strategies determine glycosylation changes at different levels (Figure 1C). For instance, de-glycoproteomics methods measure the glycosylation macro-heterogeneity change, which refers to the change in site-specific glycosylation level (can be a result of glycoprotein expression change) or glycosylation site occupancy after normalizing to the abundances of non-glycosylated peptides from the same protein. In contrast, intact glycopeptide analysis often focuses primarily on the micro-heterogeneity (i.e., the relative abundances of all detectable glycoforms on a glycosite) or the relative quantity of each unique glycoform across samples to be compared. For global glycan changes, researchers often rely on a separate glycomic analysis after releasing the glycans from glycopeptides. Integrating separate analyses of intact glycopeptides, de-glycopeptides and non-glycopeptides from the same samples is often required for generating a comprehensive picture of the underlying mechanisms [3,16,17,18,56,80]. Complicated glycosylation quantification strategies inevitably result in technical challenges for non-expert laboratories and limit their application.

We recently proposed the SugarQuant workflow for multiplexed quantitative site-specific N-glycoproteomics [27]. In the workflow, we introduced a data processing pipeline termed GlycoBinder for streamlined protein glycosylation quantification at multiple levels. As mentioned above, GlycoBinder extracted and merged TMT reporter intensities for each identified glycopeptide. We then determined the abundance of each site-specific glycoform and deduced its intensity ratio among samples to be compared (Figure 5A). Most multiplexed quantitative glycoproteomic studies stopped at this level. We took our study a step further and propagated the intensities of site-specific glycoforms to determine the glycosylation changes of individual glycosites (Figure 5B). Therefore, SugarQuant enables the simultaneous determination of the site-specific macro- and micro-heterogeneity of protein glycosylation in a single experiment. We also extended the strategy to calculate the changes in unique glycans by summing the intensities of all glycoforms bearing the same glycans (Figure 5C). To validate the SugarQuant result, we profiled the glycoproteome of nine diffuse large B-cell lymphoma (DLBCL) cell lines using the SugarQuant pipeline and a separate global de-glycopeptide analysis [110] Next, we compared the glycosylation levels of quantified glycosites determined by the two methods in the DLBCL cells. As shown in Figure 5D, our results revealed a good correlation (mean Pearson correlation coefficient > 0.7) between the two approaches. Although this concept was only validated in isobaric labeling-based glycoproteome quantification, it is in theory applicable to other quantification methods.

## 6. Applications of Quantitative Glycoproteomics

### 6.1. Cancers and Other Diseases

Glycosylation plays critical roles in protein functions and cellular processes, and its aberration is closely associated with human diseases, including cancers [111], neurodegenerative diseases [112,113], immunology disorders [49] and virus infection [114]. Quantitative glycoproteomics studies characterize disease-related micro- and macro-heterogeneous glycosylation changes in purified proteins (e.g., immunoglobulins and therapeutic proteins) [49,115,116,117,118,119,120,121,122] and complex samples, including serum [52,66,67], urine [51,123], cerebrospinal fluid [124], milk [125], tissues, cells and exosomes [126,127]. Altered protein glycosylation is a hallmark of various cancers and has become a promising target for disease biomarkers [128]. Rapidly improving quantitative glycoproteomics technologies have emerged as important tools for the targeted detection of glycoproteins and for the system-wide discovery of glycosylation regulations.

Alpha-fetoprotein (AFP) and core fucosylated AFP-L3 are closely correlated with hepatocellular carcinoma (HCC) and have been successfully used as clinical biomarkers for HCC diagnosis. The AFP level shows a huge variation among HCC patient populations. By relatively quantifying the intact N-glycopeptides between low and high levels of AFP tumor groups, several sialylated but not core fucosylated tri-antennary glycans were uniquely increased in HCC tumors with low AFP level, while many core fucosylated bi-antennary or hybrid glycans as well as bisecting glycans were uniquely increased in tumors with high AFP [22]. Other important glycoproteins with altered glycosylation were also proven to be potential biomarkers. Haptoglobin (Hp) is one of the acute-phase response proteins secreted upon liver cirrhosis. Shu et al. quantified 26 intact O-glycopeptides on Hp and found that most of them were elevated in the sera of patients with HCC compared to liver cirrhosis [129]. In addition, two intact N-glycopeptides of IgA2 (TPLTAN205ITK (H5N5S1F1) and (H5N4S2F1)) were also found to be significantly elevated in the sera from patients with HBV infection and even higher in HBV-related liver cirrhosis patients compared with healthy donors while they were then reduced in HBV-related HCC patients [43]. In addition, site-specific glycoforms of serum α-1-antitrypsin (A1AT) in early-stage HCC and cirrhosis patients showed distinct patterns between liver cirrhosis and HCC patients [71]. Comparison of the plasma proteins of healthy controls and cirrhotic patients showed an average 1.5-fold increase in the fucosylation of bi-antennary glycoforms and a 3-fold increase in the fucosylation of tri- and tetra-antennary glycoforms [64]. Glycoproteomics also helps the functional investigation of glycosylation in HCC. The dynamic alterations of site-specific glycosylation during HGF/TGF-β1-induced EMT in three HCC cell lines were systematically investigated using precision glycoproteomic methods, and the increased core-fucosylation of FOLR1 enhanced the folate uptake capacity of HCC cells to promote EMT [130].

Prostate-specific antigen (PSA) is a US Food and Drug Administration (FDA)-approved serum biomarker for prostate cancer (PCa) screening, and its glycosylation profiles in PCa patients provide the potential for improving the specificity of the PSA test [131,132]. A HILIC-MRM strategy was further developed to efficiently separate the isomeric sialic acid linkage-specific PSA glycoproteoforms, which may improve PCa diagnosis or screening [133]. To complement the prostate-specific antigen (PSA) in PCa diagnostic screening programs, urine N- and O-glycoproteomic profiling from PCa and benign prostatic hyperplasia (BPH) were compared. A panel of 56 intact N-glycopeptides that can nicely discriminate PCa and BPH were identified [16]. In a separation study, glycomic and glycoproteomic analyses of surgically removed PCa tissues spanning five histological grades (G1–G5, *n* = 10/grade) and BPH tissues (*n* = 5) revealed several protein-, cell- and tumor-grade-specific N- and O-glycosylation [17]. Quantitative glycoproteomics indicated cell-specific dynamics of pauci- and oligomannosylation during PCa progression and increased N-glycan branching and core 2-type O-glycosylation in extracellular matrix glycoproteins. To understand the underlying pathogenesis of castration resistance, the Zhang group profiled the proteome and glycoproteome of LNCaP and PC3 prostate cancer cell lines, which are models for androgen dependence and androgen independence, respectively. The increased fucosylation in PC3 cells were proven [24]. They further analyzed core fucosylated glycopeptides in non-aggressive (NAG) and aggressive (AG) prostate cancer cell lines and also detected increased fucosylation in AG cell lines compared to NAG cells [23].

Glycoproteomics-based signatures can be used for tumor subtyping and clinical outcome prediction in ovarian cancer and provide potential for precision medicine [134]. Zhang’s group first performed an integrated proteomic and glycoproteomic analysis of 83 high-grade serous ovarian carcinoma (HGSC) tumors and 23 non-tumors, which revealed tumor-specific glycosylation, uncovered different glycosylations associated with three tumor clusters and identified glycosylation enzymes correlated with glycosylation alterations [3]. Then, to further investigate the roles of protein glycosylation in the heterogeneity of HGSC, the same group further performed mass spectrometry-based site-specific glycoproteomic characterization of 119 TCGA HGSC tissues. The intact glycoproteomic profiles classified three major tumor clusters and five groups of intact glycopeptides. Intact glycopeptide signatures of the mesenchymal subtype are associated with a poor clinical outcome of HGSC [4]. A deeper understanding of the glycoproteomic signatures of HGSC may provide important clues for precision medicine and tumor-targeted therapy.

As nicely summarized by Khan and Cabral, the aberrant glycosylation of markers of cancer stem cells (CSCs) in key cellular signaling pathways has been directly correlated with the self-renewal properties and drug-resistant mechanisms [135]. The Tian group performed site-specific N-glycoproteomics to characterize the differential N-glycosylation in MCF-7 cancer stem cells relative to MCF-7 cells, leading to the discovery of potential N-glycoprotein markers of MCF-7 cancer stem cells [30]. They extended the analysis to adriamycin-resistant breast cancer stem cells for the discovery of changed glycosylation involved in the drug-resistant mechanism [31,32,33,34,35].

Protein glycosylation alteration is also associated with other cancers, including breast cancer, pancreatic cancer and gastric cancer. Fang et al. identified a group of 11 glycopeptides from serum mannose receptor as a potential marker for differentiating and stratifying breast cancer subtypes [19]. Another glycoproteomic study further revealed distinct glycosylation micro-heterogeneity in pyruvate kinase isozyme M2 (PKM2) knockout cells and parental breast cancer cells [13]. In addition, Lu et al. reported the potential of fucosylated SERPINA1 as a novel plasma marker for pancreatic cancer diagnosis based on their quantitative tissue N-glycoproteomics analyses [136]. On the other hand, a recent glycosylation profiling of the ErbB2 ectodomain, an oncogenic cell surface receptor tyrosine kinase, revealed a site-specific glycosylation profile in gastric cancer cells [137]. By specifically targeting the ErbB2 N-glycosylation sites in the trastuzumab-binding domain, ST6Gal1-mediated aberrant α-2,6-sialylation actively tunes the resistance of ErbB2-driven gastric cancer cells to trastuzumab. These studies highlighted the important roles of quantitative glycopeptide analysis in cancer biomarker discovery and the exploration of underlying mechanisms [4].

In addition to cancers, dysregulated N-glycosylation occurs in neurological and cardiac disorders. For instance, recent N-glycoproteomic studies of brains or cerebrospinal fluid obtained from patients or mice with Alzheimer’s disease revealed highly heterogeneous and dysregulated protein N-glycosylation alterations [56,124,138]. Glycoproteomic profiling of neonatal mouse hearts showed an overall upregulation of sialylation and core fucosylation during transient regeneration [53].

### 6.2. SARS-CoV-2

Severe acute respiratory syndrome coronavirus 2 (SARS-CoV-2) has caused the COVID-19 pandemic. The heavily glycosylated spike (S) protein located on the SARS-CoV-2 surface binds to human angiotensin-converting enzyme 2 (ACE2) and mediates host-cell entry [139]. Glycans carried on the S protein facilitate immune evasion by shielding specific protein epitopes from antibody neutralization [140]. MS-based quantitative glycoproteomics quickly decoded the macro- and micro-heterogeneity of N-/O-glycosylation on both the SARS-CoV-2 spike protein and human ACE2. A total of 22 N-linked glycosylation sites have been detected until now from the in-vitro-expressed S protein ectodomain and the S protein extracted from virions [139,141]. Recently, up to 27 O-glycosylation sites were identified on recombinant SARS-CoV-2 S proteins [142,143,144], and an “O-Follow-N” rule, whereby O-glycosylation occurs near the glycosylated Asn in N-sequon, was proposed [141]. These studies revealed the micro- and macro-heterogeneity of glycosylation of the SARS-CoV-2 Spike protein. Unoccupied glycosylation sites were detected on the S protein, although they were quantified as a very minor component of the total peptide pool. The high occupancy of N-linked glycan sequons of SARSCoV-2 S indicates that recombinant immunogens will not require further optimization to enhance site occupancy [139].

However, it is crucial to note that recombinant S proteins showed different site-specific glycosylation than those originating from wild-type SARS-CoV-2 virions [145]. Furthermore, distinct expression cells also produced S proteins carrying different glycosylation [143,146,147]. Such variations in glycosylation may influence the studies for developing inhibitors, antibodies and vaccines.

In addition to the S protein, its binding partner, the ACE2 receptor, is heavily glycosylated, too. All seven glycosylation sites in hACE2 were found to be completely occupied, mainly by complex N-glycans [148,149]. Interestingly, the glycans at two glycosylation sites of hACE2, N90 and N322, may have opposite effects on spike protein binding by using atomistic molecular dynamics (MD) simulations [150]. In addition, nucleocapsid protein (N protein), one of the most abundant proteins in coronaviruses, is also found to be highly N- and O-glycosylated [151]. Quantitative glycoproteomics methods have timely contributed to understanding the SARS-CoV-2 glycosylation during the COVID-19 pandemic and are undoubtedly essential for future vaccine development.

## 7. Conclusions

Glycoproteomics methods are maturing along with the ever-improving algorithms for confident and fast glycopeptide identification at the proteome scale. These methods allow researchers to uncover the glycosylation regulations and the underlying mechanisms in disease development. However, the inherent complexity of glycosylation limits the direct application of the well-developed quantitative methods for large-scale intact glycopeptide quantification. There are numerous technical considerations when planning a quantitative glycoproteomics study, such as labeling efficiency, the mass difference among the isotope-coded labels, variations introduced by glycopeptide enrichment and the available data processing tools. Distinct methods can significantly differ in quantitative performance, sensitivity, throughput and cost. This article summarizes the currently applicable approaches for quantitative glycoproteomics and discusses their advantages and disadvantages individually (Table 1).

In general, isobaric chemical labeling provides unprecedented multiplexing capability (up to 21 channels), decreasing both instrument time and run-to-run variation. Ion mobility and MS3-based reporter quantification can efficiently reduce the ratio distortion caused by co-isolation interference and improve quantitative performance. In contrast, metabolic labeling allows sample pooling at the very first step of sample preparation, avoiding technical variations introduced during sample preparation. However, limited by available isotope-coded reagents, metabolic labeling often introduces a mass shift on the labeled glycopeptides, which complicates the MS1 scans. Overlapping isotopic peaks of the labeled glycopeptides can further lead to biased quantification.

Label-free quantification enables a straightforward and cost-effective sample preparation workflow but is hungrier for instrument time and more vulnerable to run-to-run variation. DIA analysis is still in its infancy for large-scale glycopeptide quantification. It improves the missing value and reproducibility but is also more challenging for proper data analysis. On the other hand, target methods provide higher sensitivity and are the methods of choice for validating potential biomarkers and clinical applications.

We also summarize recent applications of quantitative glycoproteomics to explore the macro- and micro-heterogeneity of glycopeptides in various diseases and reveal glycosylation regulation in disease progression. Such analyses often involve the integration of multiple methods (e.g., glycomics, de-glycoproteomics, glycoproteomics, and proteomics) to construct a global view of glycosylation regulation. To ease the technical challenges, we proposed a streamlined approach for obtaining multi-layered glycosylation quantification in a single experiment. Glycopeptide analysis has also contributed to studying the glycan shield on proteins of coronavirus strains that cause the ongoing pandemic. We anticipate that rapidly developing MS technologies and software tools will bring about more accurate, precise, sensitive and user-friendly quantification methods for intact glycopeptides, accelerating biomarker discovery and therapeutic target identification.

## Figures and Tables

**Figure 1 ijms-23-01609-f001:**
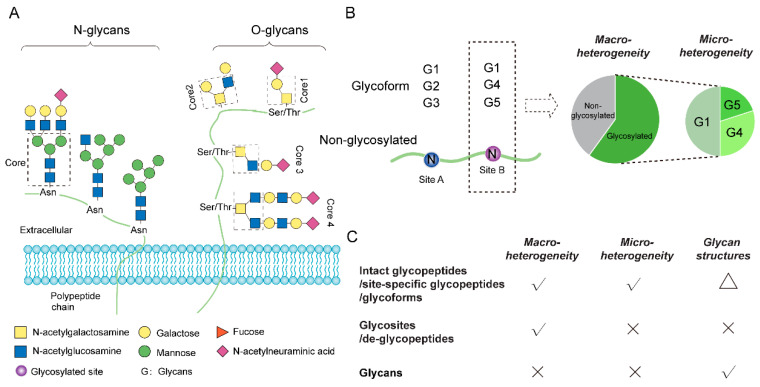
Protein glycosylation and its macro- and micro-heterogeneity. (**A**) Depicted N- and O-linked glycan structures that present on proteins. (**B**) Examples of macro- and micro-heterogeneity. Each glycosylated site on a protein may be only partially occupied by various glycans (i.e., site-specific glycoforms). Macro-heterogeneity indicates the abundance or percentage of all glycosylated forms at each site. Micro-heterogeneity represents the relative abundances of the glycoforms at each site (e.g., G1, G2, G3 at site A and G1, G4, G5 at site B). (**C**) Available information that different layers of glycosylation analysis can offer. Intact glycopeptide analysis allows quantification at the glycosite (macro-heterogeneity), glycoform (micro-heterogeneity) and glycan levels. The triangle indicates that intact glycopeptide analysis does not characterize glycosidic linkages of the glycan structure (only glycan composition).

**Figure 2 ijms-23-01609-f002:**
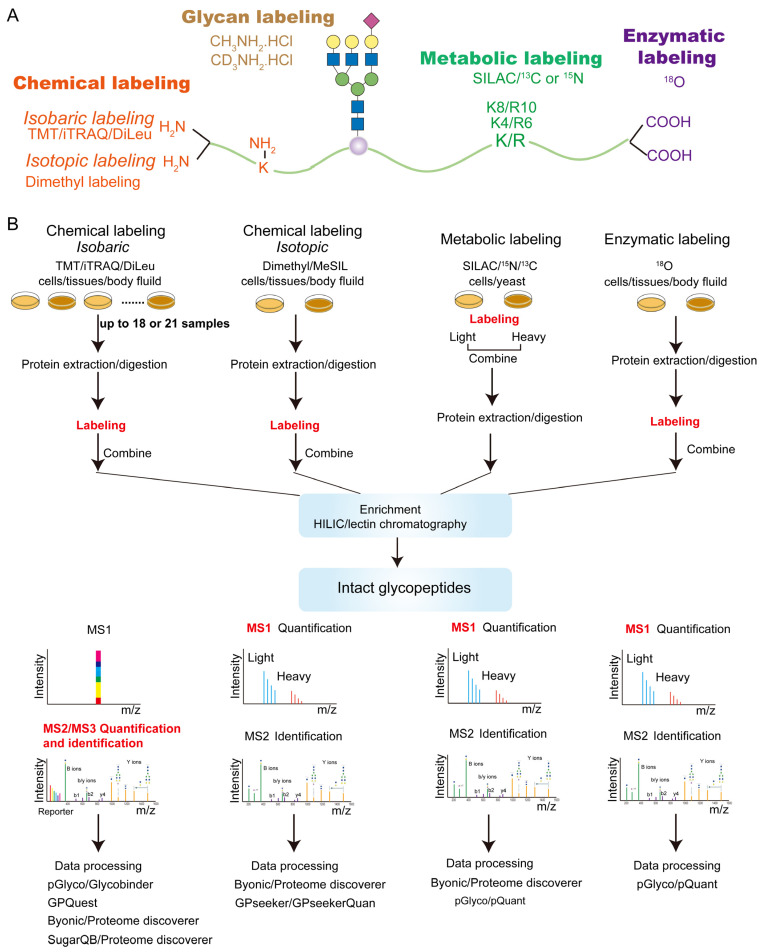
Labeling-based strategies for quantitative glycoproteomics. (**A**) Available positions on an intact glycopeptide for different labeling strategies. For example, chemical labeling reagents react with the amines at peptide N-termini or lysine side chains. (**B**) Sample preparation workflow for various labeling strategies.

**Figure 3 ijms-23-01609-f003:**
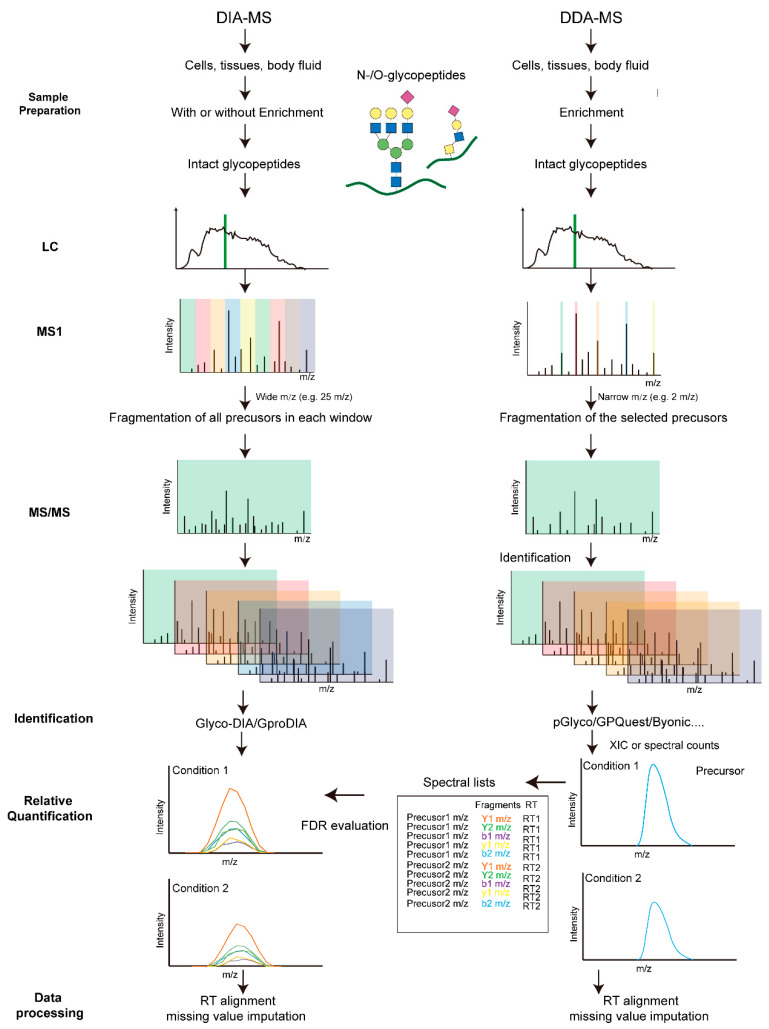
Schematic comparison of DIA- and DDA-based quantitative glycoproteomics.

**Figure 4 ijms-23-01609-f004:**
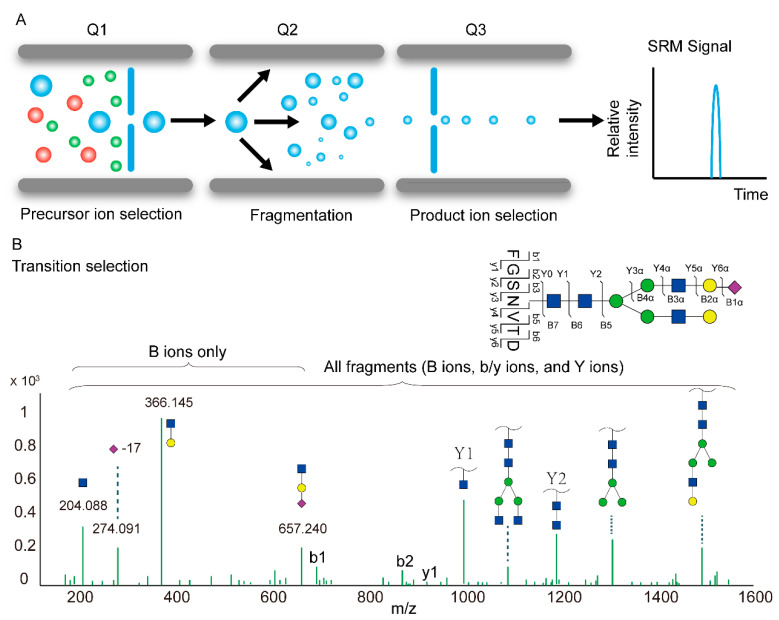
Scheme of SRM (**A**) and the strategies for selecting glycopeptide SRM transitions (**B**).

**Figure 5 ijms-23-01609-f005:**
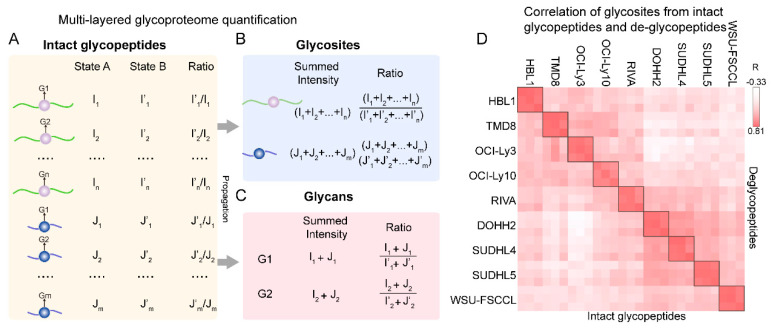
Multi-layered glycoproteome quantification. (**A**) Quantification at the intact glycopeptide level. I and J represent the intensities of reporter ions in state A and I’ and J’ represent the intensities of reporter ions in state B. The relative abundances of a glycopeptide between two states (A versus B) were obtained by comparing the intensities of their reporter ions. (**B**) Quantification at the glycosite level. The summed intensities of all glycoforms on the same glycosite (i.e., glycoforms including G1 to Gn on the purple ball, which represent the glycosite) were compared between two states. (**C**) Quantification at the glycan level. The summed intensities of all glycoforms on the same glycan (i.e., G1 on purple ball and G1 on blue ball) were compared between two states. (**D**) The correlation of glycosite quantification between the SugarQuant output and the separate quantitative de-glycoproteome experiments of the same samples from nine cell lines with three biological replicates.

**Table 1 ijms-23-01609-t001:** Overview of the strategies for quantitative glycoproteomics.

Methods	Reagent	Principle	Sample	Multiplexity	MS Level	Advantages	Disadvantages	Ref.
Isobaric chemical labeling	TMT/ iTRAQ /DiLeu/ IBT	React with amine on peptides	Cells, tissue, fluid	2, 4, 6, 10, 11, 16, 18, 21	MS2 or MS3	Enhanced signal intensity in MS and MS/MS; high multiplexing capability; simple data analysis; reduced measurement time; applicable to any sample; reduced run-to-run variations; low missing values	Expensive for commercial reagents; Does not allow in vivo labeling	[3,4,13,14,15,16,17,18,19,20,21,22,23,24,25,26,27,28]
Isotopic chemical labeling	Dimethyl/Diethyl	React with the carboxyl groups of peptides	Cells, tissue, fluid	3	MS1	Low costs; simple in handling; applicable to any sample types	Incomplete labeling complicates data analysis; side reactions; limited multiplexing capability (up to 2-plex); not suitable for in vivo labeling	[9,29,30,31,32,33,34,35,36,37,38,39]
Metabolic labeling	SILAC	Metabolic labeling with amino acids containing stable heavy isotopes when culturing cells	Cells	2 or 3	MS1	Allow in vivo labeling, minimize system errors; applicable to cells but can be expanded to tissues or model organisms using internal standards (e.g., superSILAC)	High costs; not applicable to many biological materials; limited multiplexity; complicated MS1 spectra of glycopeptides; over-sequencing of same glycopeptides	[40]
Enzymatic labeling using ^18^O stable isotope	^18^O water	Introduce ^18^O atoms into the carboxyl termini of intact glycopeptides during tryptic digestion	Cells, tissue, fluid	2	MS1	Low costs; simple in handling; applicable to any sample (cells, animal or human tissue)	Incomplete labeling complicates data analysis. Limited multiplexing capability (up to 2-plex); not suitable for in vivo labeling	[41,42,43]
Glycan labeling	^15^N/^13^C	Metabolic labeling when culturing with ^15^N or ^13^C media	yeast	2	MS1	Can be used for the evaluation of FDR of glycopeptide search engine.	Complicated data analysis	[44]
Glycan labeling	Methylamine stable isotope labeling (MeSIL)	Label the carboxyl groups on both the sialic acid and the peptides	Cells, tissue, fluid	2	MS1	Label intact N-glycopeptides by one-step reaction easily with high labeling efficiency; distinction of neutral and sialylated glycopeptides	No description	[45]
DDA-based LFQ	XIC/ intensity	XIC or intensity of glycopeptides across runs	Cells, tissue, fluid	No limited sample numbers	MS1	No labeling required; applicable to any sample; simplified sample handling;	Huge variations in replicate measurements; longer data acquisition time; requires more computationally sophisticated data analysis; severe missing values	[46,47,48,49,50,51,52,53,54,55,56]
DDA-based LFQ	Spectra counts	The number of identified glycopeptide spectra matches	Cells, tissue, fluid	No limited sample numbers	MS1	No labeling required; applicable to any sample types; simplified sample handling;	Requires large sample size (spectral counts) to confidently predict small changes in expression; lower accuracy than labeling and XIC-based LFQ methods; severe missing values	[57,58]
DIA-based LFQ	DIA-label free	XIC of glycopeptides	Cells, tissue, fluid	No limited sample numbers	MS1	No labeling required; applicable to any sample types; simplified sample handling; higher sensitivity, reproducibility and less missing values than DDA;	Needs constructing the sample specific glycopeptides spectra libraries	[59,60,61,62,63,64,65,66,67]
Target analysis	SRM /MRM /PRM	Monitor the target precursor and product ions	Cells, tissue, fluid	No limited sample numbers	MS1	Very high sensitivity, reproducibility	The number of precursor ions to be monitored is limited by the scan speed of MS	[68,69,70,71,72,73]

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
