# Peer review of "Strategies for Proteome-Wide Quantification of Glycosylation Macro- and Micro-Heterogeneity"

_ijms, 2022, doi:10.3390/ijms23031609_

Round 1
Reviewer 1 Report
The manuscript presented by Fang P., et al. provides a decent literature review of currently existing quantitation strategies to assess glycoprotein glycoform heterogeneity in highly complex samples. The authors provide a balanced overview of current key publications in the field, clarify differences in the respective analytical concepts used and conclusively discuss the different strategies in the field. The manuscript is elegantly written and an agreeable read! I recommend publication of this work after having addressed only very minor comments from my side.
Minor remarks:
line 188-189 - "After enrichment, differently labeled glycopeptides were co-elute chromatographically and appeared distinguishable in the MS1 spectra." I recommend removing "were".
line 190-191 - "The relative peak intensity/area of the light and heavy-labeled glycopeptides were com-pared." "are"?
The authors use "e.g.," or "i.e.," at numerous positions in the text. Not being a native speaker i find this style somewhat awkward. I recommend removing the commas after the period.
line 217-219 - "Zhang et al. used differential glycan oxidation and demethylation-based quantitative de-glycopeptide analysis to determine N-sialoglycan occupancy rates on glycoproteins
[50]". - It should read "DImethylation".
line 479-481 - "SRM analysis was first developed with a triple quadrupole, where the first quadru-pole (Q1) scans for a pre-defined precursor, which undergoes CID fragmentation in the second quadrupole (Q2)." - Here, I suggest to rephrase to "SRM ananlysis was initially developed on triple quadrupole instruments,[...]".
line 482-483 - "However, such targeted analyses
have not been adapted to most current tandem mass spectrometers." - This statement is not clear. Please change or elaborate.
line 492-493 - "[...]and quantify the fucosylated
hemopexin and complement factor H in the plasma of patients with liver disease [...]" - I suggest "the fucosylated fraction of hemopexin".
line 503-505 - "N-linked glycopep-tides typically display heterogeneous glycoforms that elute a narrow retention time win-dow, resulting in overlapped LC peaks." - Again, not being a native speaker, this reads somewhat strange to me. I suggest to use "elute (WITH)IN" and "resulting in overlappING peaks."
Reviewer 2 Report
This review summarizes recent developments in mass spectrometry-based glycoproteomics quantification strategies that allow the observation of both macro- and micro-heterogeneity of protein glycosylation. Examples of biomarker discovery for human diseases and quantification of glycoproteomics in therapeutic targets are also introduced.
The manuscript was well-organized and written in general. However, it is necessary to clarify and refine the content for a better manuscript.
1) It would be good to change the title of Part 1 to cover all the contents of the chapters.
ex) Stable isotope labeling-based quantification → Labeling-based quantification
2) In order to improve the understanding of Figure 2, it is necessary to clearly divide and rearrange the sections in the figure.
3) It seems necessary to mention the attempts to improve the labeling efficiency of glycopeptide compared to peptide analysis in Line #120.
4) Since there is only one proposed application for metabolic labeling (1.3), further reinforcement of the case is necessary.
5) There is no explanation of pGlyco/pQuant in the main body 1.1 isobaric chemical labeling part and 1.3 metabolic labeling part.
6) In Table 1, the number of Isotopic chemical labeling – Multiplexity is 2, but it is described as up to three in the text line 186, so it needs to be corrected.
7) Regarding the presentation of GlycoBinder, a data processing pipeline developed by authors, additional explanation is needed in the text and in the caption of Figure 5 to help the reader understand.
8) Text lines 463-474: The summary of DDA, DIA, and LFQ is unnatural and awkward and needs clarification. For example, a context (line 470-471) is considered a description of a DIA, not a DDA. overall.
